# Turner Syndrome and the Thyroid Function—A Systematic and Critical Review

**DOI:** 10.3390/ijms252312937

**Published:** 2024-12-02

**Authors:** Katarzyna Lacka, Nikola Pempera, Alicja Główka, Agnieszka Mariowska, Miłosz Miedziaszczyk

**Affiliations:** 1Department of Endocrinology, Metabolism and Internal Medicine, Poznan University of Medical Sciences, 60-355 Poznan, Poland; 2Students’ Scientific Society, Poznan University of Medical Sciences, 61-701 Poznan, Poland; 3Department of General and Transplant Surgery, Poznan University of Medical Sciences, 60-355 Poznan, Poland; m.miedziaszczyk@wp.pl; 4Department of Clinical Pharmacy and Biopharmacy, Poznan University of Medical Sciences, 60-806 Poznan, Poland

**Keywords:** Turner syndrome, thyroid function, autoimmunity, thyroid hormones, anti-thyroid antibodies, hypothyroidism

## Abstract

Turner syndrome (TS) is associated with thyroid disorders. Since the rate of thyroid disease among patients with this syndrome is significantly higher as compared to the general population, it seems vital to explore this particular area. This systematic and critical review was performed to evaluate thyroid function and autoimmunity in patients with Turner syndrome. Four databases were searched: PubMed, Scopus, Google Scholar, and Cochrane Library from the onset of the study to July 2024. Two independent researchers manually searched databases for the following keywords: “Turner syndrome”, “anti-TPO”, “anti-Tg”, “autoimmune thyroid disorders”, “TSH”, and “hypothyroidism”, which were entered into the search engine in isolation, as well as in combinations. Criteria incorporating information on thyroid-stimulating hormone (TSH), triiodothyronine (total—TT3), and thyroxine (free and total—fT4, TT4) concentrations among patients and control groups were also included. Thyroid diseases are common in patients with Turner syndrome. Women with TS present both higher TSH levels and positive thyroid antibodies concentrations. Typical thyroid ultrasound heterogeneity with a hypogenic or mixed echopattern was also observed. As a result, it is essential to monitor thyroid hormone levels in this group, in order to detect hypothyroidism earlier and initiate appropriate replacement therapy. Thyroid diseases in women with TS may remain underdiagnosed for a number of years, due to the lack of screening. Therefore, the authors suggest a thyroid screening regimen for TS patients, which allows for early detection of the disease and implementation of treatment.

## 1. Introduction

Turner syndrome (TS) is a genetic disease characterized by quantitative and/or structural aberrations in one of two X chromosomes with frequent mosaicism. According to different sources, TS incidence is estimated between 1 in 2000 to 1 in 2500 live female births [1,2]. Approximately 1.5 million women worldwide suffer from TS [3].

X aberrations comprise X monosomy (45,X) and mosaicism (50–70%), which includes 45X/46,XX, 45,X/46XY, 45,X/46,XX/47,XXX, 45,X/46X,i(Xq), 45,X/46, and Xdel(Xp), as well as structural X aberrations, such as the total or partial deletion of the short arm of an X chromosome (46,Xdel(Xp)), the isochromosome of the long arm of an X (46,X,i(Xq), ring chromosome (46,X,r(X)), and marker chromosome (46,X + m) [4]. To date, no correlation has been found between the onset of TS and the age of the parents, environmental factors, or the use of specific drugs/toxins. Additionally, Turner syndrome is observed in all races all over the world, and it is not hereditary; therefore, it can occur even in families with healthy individuals [5,6].

TS was first named in 1938 by H.H. Turner, who was a physician in Oklahoma [7]. He described seven cases of women with short stature, webbed neck, valgus elbows, low neck hairline, and sexual infantilism. Notably, however, the first description of TS, albeit still unnamed at the time, dates back to 1760, when G.B. Morgagni performed an autopsy on a monk whose aorta was markedly narrowed near the heart [8].

Women with Turner syndrome show typical phenotype features, including short stature; a short and webbed neck; cubitus valgus; a broad chest with widely spaced nipples; and lymphedema of the hands and feet, which is particularly common at birth. Additionally, the characteristics of TS comprise low positional and dysplastic ears, antimongoloid slant, epicanthus, poor facial expression, low descending hairline on the neck, gothic palate, and micrognathia [6,9,10].

Individuals with Turner syndrome (TS) are prone to develop autoimmune disorders, particularly thyroiditis, the most frequent of which is Hashimoto’s disease [11]. Thyroid autoimmune diseases are characterized by abnormal lymphocytic activation, directed against self-antigens. The cause of autoimmune diseases in TS remains unclear [11], although there are several mechanisms that suggest a potential correlation between the X-isochromosome and increased prevalence of anti-thyroid peroxidase (TPO-Ab) antibodies. Other researchers, in turn, suggested that a genetic factor located on the X-gene could play a huge role in the development of thyroid autoimmune diseases [12], a claim that was later confirmed by another large study [13]. Nonetheless, the risk of hypothyroidism or hyperthyroidism in Turner syndrome remains independent of the karyotype [14,15], and the treatment goals do not differ from those established for healthy populations [12,14].

The frequency of signs and symptoms varies in different patients with Turner syndrome and may depend on the type of X chromosome aberrations [16,17,18,19,20].

The largest number of immune-related genes is housed in the X chromosome [21], where genes FOXP3 and PTPN22 are presumed to contribute most commonly to the pathogenesis of TS [22]. In fact, the presence of two X chromosomes and skewed inactivation of one of them are thought to play a protective role in females, who generally present a stronger immune response to diverse pathogens. Nevertheless, the hyperresponsiveness of the immune system in females results in a predisposition towards auto-immune diseases [23]. In addition, autoimmune diseases are 2–3 times more frequent in women with Turner syndrome [24]. Women suffering from TS present at higher-than-average risk of developing autoimmune thyroid diseases, where hypothyroidism appears most commonly, mainly in the subclinical form, occurring in about 58% of TS women. It may develop not only in adolescence and adult age, but also in childhood [25,26], and is characterized by elevated TSH levels, while maintaining normal T4 concentrations [27]. Scientific literature also provides a case of Hashimoto’s encephalopathy co-existing with Turner syndrome [28], which is a rare disease supposed to have an autoimmune pathogenesis [29]. Nevertheless, up to now, the autoimmune mechanism underlying the abovementioned syndrome has not been completely understood and, thus, it requires further research.

Due to a higher incidence of autoimmune diseases in patients with TS, the current guidelines recommend screening for hypothyroidism at the time of the diagnosis using TSH and a fT4 (free-thyroxine), which should commence in early childhood and be subsequently performed annually. However, there are no recommendations for the routine testing of thyroid antibodies [2].

In view of the above, the aim of our study was to evaluate the association between TS and autoimmune thyroid diseases.

## 2. Results and Discussion

### 2.1. Characterization of Considered Population

Our systematic and critical review comprised eight articles in total, and each of them included a study and control group. Control groups consisted exclusively of healthy women. Subjects in both groups did not differ significantly in terms of age, although significant differences were observed for height and weight in the TS population as compared to the healthy individuals. In contrast, body mass index (BMI) was increased in patients with Turner syndrome. The relation between these parameters is demonstrated in Table 1.

In the study by Mitsibounas et al. [30], the study group included 15 of 29 patients who were administered only hormones (estrogen and progestogen). A similar approach was adopted by Wasniewska et al. [31]—in their study group, 29 out of 66 women received estrogen therapy. In turn, Susperreguy et al. [32] implemented growth hormone (GH) therapy in 10 out of 20 women suffering from TS and described the observed results following GH treatment. Administration of the hormone at a dose of 0.33 mg/kg/week for a period between 6 months and 2 years resulted in a decrease in TT4 as well as fT4 and, simultaneously, an increase in TT3 and TSH serum levels. In turn, in the study by Naessen et al. [26], 239 of 503 from the study group received growth hormone and 377 of 503 received menopausal hormone therapy. Interestingly, they observed no differences in the prevalence or incidence of autoimmune disorders.

### 2.2. Thyroid Parameters

Individuals with Turner syndrome (TS) are prone to developing autoimmune disorders, in particular Hashimoto’s thyroiditis [11]. They are characterized by abnormal lymphocytic activation, directed against self-antigens, such as thyroglobulin (Tg) and thyroperoxidase (TPO) [37]. This was first observed in 1948, when the postmortem of a female TS patient revealed a lymphocytic infiltration of a small thyroid gland [38]. In a healthy population, the diagnosis of thyroiditis is based on clinical evidence and symptoms of thyroid dysfunction. In terms of TS, the functional examination is performed periodically, regardless of the clinical picture, which allows for the detection even of subclinical changes [33]. Gravholt et al. in their study demonstrated that the genes IL3RA and CSF2RA located on the X chromosome are differentially methylated in women with Turner syndrome in comparison to healthy women with a karyotype of 46,XX. Furthermore, they hypothesized that IL3RA could be associated with an increased risk of autoimmune diseases in Turner syndrome, such as thyroiditis [39].

It is worth bearing in mind that TS should always be differentiated from Noonan syndrome (NS). Although the conducted studies reported the occurrence of thyroid antibodies in NS, hypothyroidism was equally common as in the normal population [40,41].

#### 2.2.1. Thyroid Function

Our systematic review analyzed thyroid function in patients with TS, which included both thyroid hormones and TSH. The data are presented in Table 2. According to the analyzed sources, TS patients presented significantly higher levels of TSH and TT3; however, no significant differences in fT4 levels were observed. This, in turn, indicates that TT3 may also constitute a useful parameter in screening patients with TS. Nevertheless, TT3 is not a perfect parameter, since it can be also elevated due to other physiological processes, such as pregnancy [42].

Several studies suggest a greater incidence of hypothyroidism in women suffering from TS [43,44]. Notably, however, a transition from one clinical phenotype to another over time may also occur [45]. We analyzed the parameters of patients who were included in eight of our studies and concluded that in the course of the study the majority of subjects were euthyroid, as presented in Table 3. Nonetheless, there is no clear evidence whether the euthyroid patients used levothyroxine at the time of the study or not. According to the sources, subclinical hypothyroidism was diagnosed on the basis of elevated serum TSH with normal total T4 and T3 and no clinical signs of hypothyroidism [26,30,31,32,33,34,35,36], whereas the clinical and overt hypothyroidism were reported when total and free T4 and T3 were low and TSH was high [26,30,31,32,33,34,35,36].

#### 2.2.2. Thyroid Antibodies

A study conducted on a group of Japanese women with TS revealed that more than half of the patients showed thyroid autoantibodies in adulthood. This, in turn, demonstrates that monitoring of thyroid hormone is vital in that particular group, in order to detect hypothyroidism earlier and implement appropriate replacement therapy [46]. In our systematic review, we also reached similar conclusions. Patients with TS more frequently presented higher antibody levels. Based on the sources [26,30,31,32,33,34,35] containing relevant data, we calculated that 342 of 806 (42.43%) patients in the study groups presented positive thyroid antibodies, whereas in the control groups, they were present only in 180 of 887 (20.29%) individuals. The difference is shown in Figure 1. The abovementioned data indicate that thyroid antibodies are more frequently observed in patients with TS than in the healthy population. Table 4 shows the differences in thyroid antibodies in patients with TS and the healthy controls.

#### 2.2.3. Thyroid in Ultrasonography

The patients in the relevant studies underwent ultrasonography. Calcaterra et al. [36] reported typical thyroid ultrasound heterogeneity with a hypogenic or mixed echopattern. Thyroid volume was significantly larger in patients with TS in comparison to the healthy population. Nonetheless, Lacka et al. [35] reached different conclusions in their research. Table 5 provides ultrasonography findings described in the sources. In general, ongoing immunological processes and the patient’s growth affect the volume of the thyroid gland. From this perspective, the relation between GH treatment and thyroid volume seems to be of importance, as potentially, GH-treated female patients should present with larger thyroid volumes. Consequently, further studies need to be conducted, in order to investigate if there are statistically significant differences in ultrasonography parameters in women treated with GH and with other hormones.

### 2.3. Screening and Treatment

According to the conducted studies, administration of the growth hormone in GH-deficient patients, such as TS women, resulted in various disorders, including changes in thyroid function. Case reports describe such dysfunctions as a decreased sensitivity of thyrotropin to thyrotropin-releasing hormone stimulation, induction of hypothyroidism, and enhanced peripheral conversion of thyroxine to triiodothyronine; however, most studies were casuistic or uncontrolled in character [46].

#### Thyroid Gland

Hypothyroidism represents one of the most common endocrine disorders. Clinical manifestations vary significantly, thus, some patients experience mild unspecific symptoms, such as tiredness, cold intolerance, lack of vitality, and obstipation, whereas others develop myxedema, which is clinically manifested as an increased sensitivity to pharmacotherapy, confusion, areflexia, megacolon, and may even be fatal [47].

Hypothyroidism reflects reduced thyroid function. It may result from the dysfunction of the thyroid gland (primary), alternatively, it may occur due to defects along the hypothalamic/pituitary axis, or stem from the intake of lower-than-required doses of exogenous thyroid hormone in patients diagnosed with primary hypothyroidism. There are two different types of hypothyroidism. The first is referred to as overt—it is diagnosed in cases where the serum thyrotropin level is greater than the upper normal limit, while simultaneously the serum free thyroxine (T4) values are below the normal range. The second type is subclinical, and it is defined as a condition where thyrotropin values are higher than the upper normal limit, with serum free T4 concentrations remaining within the reference range [48]. Nevertheless, the association between the subclinical form and the development of symptoms remains uncertain [49].

Some researchers indicate that patients with a subclinical form tend to develop symptoms of depression more frequently [50].

Most guidelines indicate treatment should be initiated when TSH levels exceed >10 mIU/L [51]. The European Thyroid Association (ETA) guidelines recommend treatment in more severe forms, while, in milder forms, levothyroxine administration should be considered in patients with repeated measures of TSH between 5–10 mIU/L, and symptoms consistent with hypothyroidism. However, if a symptomatic response is not achieved within 3–4 months following TSH normalization, treatment should be discontinued [52]. As symptoms are unspecific, the treatment decision in subclinical hypothyroidism should be individualized [53], taking into consideration factors, such as age, pregnancy, smoking, weight, and ethnicity [54]. Moreover, the risks and potential benefits need to be assessed in each case. Notably, if untreated, overt hypothyroidism may have multi-organ consequences, including the cardiovascular system [48,55]; whereas, treatment of the subclinical form may result in thyrotoxicosis [56]. Nonetheless, the treatment goals remain the same as in other populations [51,57,58].

Due to the significant risk of thyroiditis in patients with TS, appropriate monitoring is essential in order to detect the disease as early as possible and to initiate treatment. According to the National Care Protocol [57], anti-TPO Ab and TSH ± fT4 should be measured at the time of diagnosis in patients older than four years of age, and subsequently the test should be repeated annually. Thyroid ultrasound should be performed when abnormalities in thyroid parameters (such as TSH, thyroid hormones, anti-thyroid antibodies) are observed. In the case of hypothyroidism, L-thyroxine replacement therapy should be introduced, and thyroid parameters should be tested every 6 months in childhood and every 12 months in adulthood. Antithyroid drugs may be used in cases of Graves’ disease [57].

### 2.4. Future Prospects

Given the karyotype distribution, congenital anomalies as well as comorbidities, patients with TS require close and complex medical surveillance throughout their entire lives.

Our systematic and critical review highlights the fact that thyroid diseases are common in the TS population [10]. Screening for hypothyroidism with measurement of TSH is recommended every 1 year, starting at 2 years of age and continuing through adulthood, as well as in cases when new symptoms appear. If TSH is increased, testing for anti-thyroid antibodies seems to be warranted [58].

Many of the discussed comorbidities associated with Turner syndrome appear to have a genetic basis of (at least partially) known etiology. Tests should include mutations which affect, or have a high probability of affecting, the development of diseases, such as: TIMP1 and TIMP3 [59], SHOX [42,60], CLTRN [61,62], IL3RA [39], CSF2RA [39], KDM6A [63] and BDNF [64].

Moreover, it is crucial to conduct further research into potential mechanisms involved in the development of hypothyroidism in patients with Turner syndrome and to perform larger-scale placebo-controlled trials. Figure 2 shows the proposed thyroid management for patients with TS that should facilitate early detection of thyroid diseases and, consequently, the initiation of treatment in the early stages of the disease.

### 2.5. Limitations

The main limitation of our analysis involves the lack of data on serum-free triiodothyronine levels, which should also be tested in patients suffering from TS. Additionally, there is no differentiation in terms of the number of patients who showed positive anti-TPO and anti-Tg levels. The sources only report the total number of antithyroid-positive patients, as well as average antibody levels.

## 3. Materials and Methods

### 3.1. Search Strategy

The presented review is designed following the PRISMA 2020 guidelines (Appendix A). A search of PubMed, Scopus, Cochrane Library, and Google Scholar was conducted throughout July 2024. The search terms included “Turner syndrome”, “anti-TPO”, “anti-Tg”, “autoimmune thyroid disorders”, “TSH”, and “hypothyroidism”, which were used as isolated entries, as well as entered in various combinations. The first selection of research papers was based on the study title and the keywords. Overall, 1936 papers were found, which had been published since the beginning of TS research until July 2024. Such a broad time span stems from the fact that even the oldest manuscripts were significant for obtaining as much data as possible. The country of origin and form of the publications were irrelevant. Two analysts, working independently, verified the search engine results to select the most relevant studies. In total, the presented systematic review comprised 8 studies, and the study protocol was registered (ID: CRD42023471448).

### 3.2. Inclusion and Exclusion Criteria

In order to meet the inclusion criteria for our review, a given research paper had to include the mean and standard deviation for the parameters listed above. Moreover, the values had to be provided for both a study group, which was represented by female patients with Turner syndrome, and a control group, which consisted of healthy women.

In addition, our review incorporated laboratory findings regarding serum thyroid peroxidase antibodies (anti-TPO) and thyroglobulin antibodies (anti-TG), as well as thyroid function including thyrotropin (TSH), free thyroxine (fT4), total thyroxine (TT4), and total triiodothyronine (TT3) in female TS patients. We also included thyroid ultrasound findings, i.e., echogenicity and volume, as well as the diagnosis of Hashimoto’s disease, hypothyroidism, and hyperthyroidism.

### 3.3. Data Extraction and Quality Assessment

Two researchers independently extracted the following data from each paper: the year of publication, the study and control group size, types and results of studies performed, and clinical status of thyroid patients. Data were extracted using Microsoft Office Excel. The results are shown in Figure 3.

Bias assessment was conducted using the RoB 2 tool to determine the validity of the data collected from the studies. Two researchers independently answered the signaling questions for each paper. The risk of bias was used to draw conclusions from the accessed data. The results are presented in Figure 4.

## 4. Conclusions

Thyroiditis remains one of the most common comorbidities in patients with TS. Therefore, regular thyroid monitoring is vital in this group of patients. Since preventive screening is rarely performed, a large number of patients remain undiagnosed, or receive treatment only at a late stage of the disease. Therefore, we suggested a thyroid screening regimen in patients suffering from TS, which would allow for early detection of the disease and the implementation of relevant treatment. Simultaneously, we emphasize the significance of the genetic factors in the prevention of comorbidities, as it appears that gene mutations may account for their presence in TS women. However, further randomized studies are essential to conclusively determine whether genes play a role in the development of thyroid diseases. The abovementioned studies would be instrumental in identifying the parameters that predispose TS female patients to the disease, thus making it possible to perform active screening and to mitigate risk factors at an early stage. Consequently, the risk of the clinical consequences of hypothyroidism in this group of patients could be diminished, improving the quality of patients’ lives.

## Figures and Tables

**Figure 1 ijms-25-12937-f001:**
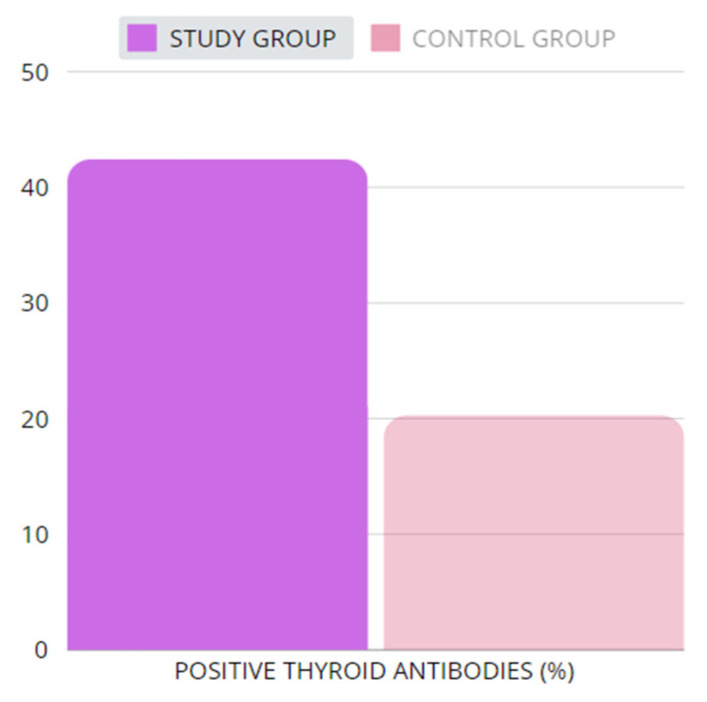
The percentage of patients with positive thyroid antibodies. The study group for the purpose of our review was calculated by means of adding the study groups from the relevant sources [26,30,31,32,33,34,35]; the same procedure was adopted for the control group. We excluded source [36] from this analysis, due to the lack of information regarding patients with positive thyroid antibodies. In total, 806 patients were included from the study groups (patients with TS), where in 342 cases, positive thyroid antibodies were found. The control group consisted of 887 individuals in total, among whom 180 presented with positive thyroid antibodies. The sources [26,30,31,32,33,34,35] do not provide a division according to antibody types, and hence there is no information if the anti-TPO and/or anti-Tg were elevated in patients. Table 4 presents levels of thyroid antibodies.

**Figure 2 ijms-25-12937-f002:**
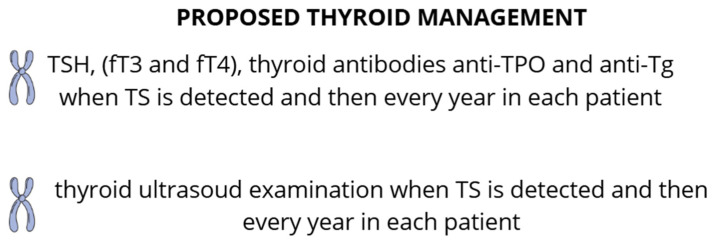
Proposed thyroid management for patients with TS.

**Figure 3 ijms-25-12937-f003:**
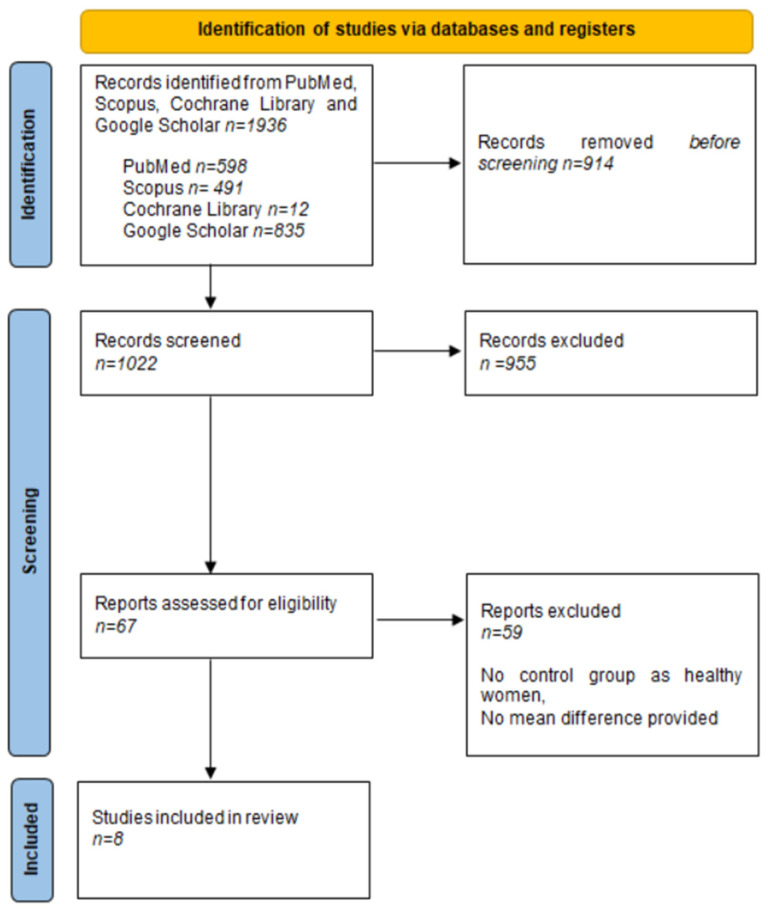
PRISMA flowchart. We performed a PubMed, Scopus, Cochrane Library, and Google Scholar search throughout July 2024. The oldest articles were also significant in order to obtain as much data as possible. Search terms included “Turner syndrome”, “anti-TPO”, “anti-Tg”, “autoimmune thyroid disorders”, “TSH”, and “hypothyroidism”, used as isolated keywords and in combination. The flowchart was performed in order to obtain information about thyroid function in women with Turner syndrome.

**Figure 4 ijms-25-12937-f004:**
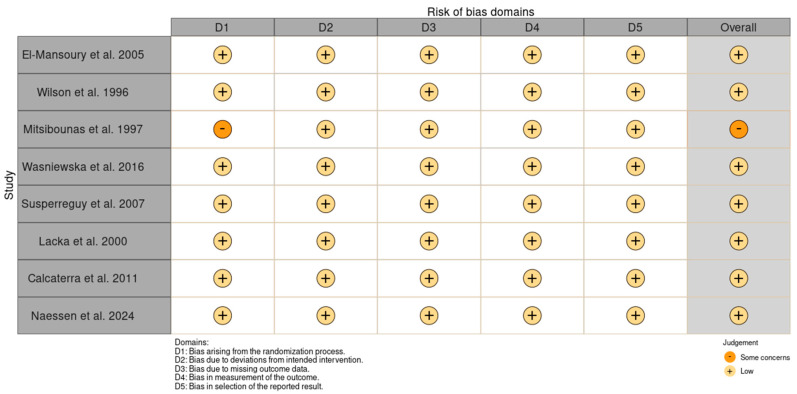
Bias assessment using version 2 of the Cochrane risk-of-bias tool for randomized trials (RoB 2) [26,30,31,32,33,34,35,36].

**Table 1 ijms-25-12937-t001:** Presentation of cases analyzed in sources [26,30,31,32,33,34,35,36]. The table includes the number of study and control groups, information regarding Hashimoto diagnosis in the analyzed groups, age, height, weight, and calculated body mass index (BMI) characterizing each group.

Author and Year		El-Mansoury et al., 2005 [33]	Wilson et al., 1996 [34]	Mitsibounas et al., 1997 [30]	Wasniewska et al., 2016 [31]	Susperreguy et al., 2007 [32]	Lacka et al., 2000 [35]	Calcaterra et al., 2011 [36]	Naessen et al., 2024 [26]
**Group (n)**	study	91	60	29	66	20 (10 treated with GH)	37	73	503
	control	228	57	22	132	10	37	93	401
**Hashimoto diagnosis (n)**	study	No data	3	No data	66	No data	No data	32	No data
	control	No data	No data	No data	132	No data	No data	0	No data
**Age (y)**	study	37.7 ± 11	No data	22.55 ± 6.34	13.0 (4.5–17.9)	9.6 ± 3.5 (TS) 10.9 ± 3.1(GH)	13.8 ± 6	25.01 ± 10.43	27.6 ± 11.5
	control	37.3 ± 9	No data	22.05 ± 2.299	10.6 (2.8–15.0)	9.9 ± 2.8	No data	1.95 to 37.2	35.5 ± 5.7
**Height (cm)**	study	149.4 ± 7	No data	148 ± 9	No data	No data	No data	No data	153.6 ± 7.2
	control	165.7 ± 6.6	No data	158 ± 5.5	No data	No data	No data	No data	166.3 ± 6.6
**Weight (kg)**	study	56.4 ± 12.4	No data	53.16 ± 10.12	No data	No data	No data	No data	58.9 ± 11.9
	control	65.4 ± 10.3	No data	52.86 ± 4.663	No data	No data	No data	No data	65.2 ± 10.2
**Body mass index (BMI) (kg/m^2^)**	study	25.3 ± 4.7	No data	23.75 ± 3.64	No data	No data	No data	No data	25.1 ± 5.7
	control	23.8 ± 3.7	No data	21.90 ± 1.681	No data	No data	No data	No data	23.6 ± 3.4

**Table 2 ijms-25-12937-t002:** Parameters according to sources [26,30,31,32,33,34,35,36].

Author and Year		El-Mansoury et al., 2005 [33]	Wilson et al., 1996 [34]	Mitsibounas et al., 1997 [30]	Wasniewska et al., 2016 [31]	Susperreguy et al., 2007 [32]	Lacka et al., 2000 [35]	Calcaterra et al., 2011 [36]	Naessen et al., 2024 [26]
**Group (n)**	study	91	60	29	66	20 (10 treated with GH)	37	73	503
	control	228	57	22	132	10	37	93	401
**Serum TSH (U/mL)**	study	2.5 ± 2.3	2.5 ± 2.5	2.468 ± 1.2003	2.8 (0.6–4.8)	2.5 ± 0.6	3.9 ± 6.61	No data	3.18 ± 7.78
	control	1.2 ± 0.7	2.4 ± 1.7	1.95 ± 1.45	2.5 (0.47–4.9)	2.8 ± 1.0	No data	No data	1.10 ± 0.63
**Serum fT4 (ng/dL)**	study	1.07 ± 0.2	No data	No data	1.173 ± 0.225	1.38 ± 0.23	1.71 ± 2.1	No data	1.227 ± 0.326
	control	1.16 ±0.2	No data	No data	1.196 ± 0.28	1.45 ± 0.28	No data	No data	1.158 ± 0.194
**Serum TT3 (nmol/L)**	study	No data	No data	1.59 ± 0.35	No data	1.7 ± 0.3	2.6 ± 0.1	No data	No data
	control	No data	No data	1.88 ± 284	No data	1.8 ± 0.3	No data	No data	No data
**Serum TT4 (nmol/L)**	study	No data	104 ± 25	124.131 ± 24.3063	No data	128 ± 23	119.7 ± 46.3	No data	No data
	control	No data	129 ± 44	119.37 ± 15.773	No data	128 ± 25	No data	No data	No data

**Table 3 ijms-25-12937-t003:** Hypo-/hyper/euthyroidism in patients with Turner syndrome [26,30,31,32,33,34,35,36].

Author and Year		El-Mansoury et al., 2005 [33]	Wilson et al., 1996 [34]	Mitsibounas et al., 1997 [30]	Wasniewska et al., 2016 [31]	Susperreguy et al., 2007 [32]	Lacka et al., 2000 [35]	Calcaterra et al., 2011 [36]	Naessen et al., 2024 [26]
**Group (n)**	study	91	60	29	66	20 (10 treated with GH)	37	73	503
	control	228	57	22	132	10	37	93	401
**Euthyroidism (n)**	study	No data	60	No data	66	20	27	73	No data
	control	No data	57	No data	132	10	No data	93	No data
**Hyperthyroidism (n)**	study	3	0	0	0	0	1	0	No data
	control	0	0	0	0	0	No data	0	No data
**Hypothyroidism (n)**	study	23	0	0	0	0	8	0	37
	control	5	0	0	0	0	No data	0	0

**Table 4 ijms-25-12937-t004:** Thyroid antibodies in patients with TS [26,30,31,32,33,34,35,36].

Author and Year		El-Mansoury et al., 2005 [33]	Wilson et al., 1996 [34]	Mitsibounas et al., 1997 [30]	Wasniewska et al., 2016 [31]	Susperreguy et al., 2007 [32]	Lacka et al., 2000 [35]	Calcaterra et al., 2011 [36]	Naessen et al., 2024 [26]
**Group (n)**	study	91	60	29	66	20	37	73	503
	control	228	57	22	132	10	37	93	401
**Positive thyroid antibodies (n)**	study	25	18	4	66	0	23	No data	206
	control	0	1	0	132	0	6	No data	41
**Serum antiTPO level (mIU/L)**	study	475 ± 960	No data	No data	119 (5–6.4)	No data	10.9 ± 12.7	No data	No data
	control	<0.01	No data	No data	374 (31–29.95)	No data	No data	No data	No data
**Serum antiTg level (mIU/L)**	study	No data	No data	No data	230 (5–4.9)	No data	3300 ± 9100	No data	No data
	control	No data	No data	No data	227 (10–6.4)	No data	No data	No data	No data

**Table 5 ijms-25-12937-t005:** Difference in thyroid volume in TS patients in comparison to healthy individuals [13,41].

Author and Year		Lacka et al., [35]	Calcaterra et al., 2011 [36]
**USG volume (cm^3^)**	study	11.03 ± 5.0	7.1 ± 2.4
	control	16.98 ± 9.1	6.1 ± 2.2
**Abnormal echogenicity (n)**	study	17	66
	control	No data	No data

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
