# Peer review of "Turner Syndrome and the Thyroid Function—A Systematic and Critical Review"

_ijms, 2024, doi:10.3390/ijms252312937_

Round 1
Reviewer 1 Report
Comments and Suggestions for Authors
Int J Mol Science 3293855
The authors have summarized, in a nice review, the literature of Thyroid function in women with Turner syndrome. The authors conclude that it is important to check the blood samples for the thyroid function annually as well as perform an ultrasound of the thyroid gland. The information is not new and is since more than 20 years included in the international guidelines. Anyhow, the review is of interest but could be condensed to focus on Hypothyroidism, or at least thyroid function, which is the main autoimmune disease in Turner syndrome. Chapter about lipids, bone and other diseases could be deleted.
A major concern is that Hypothyrodism is not defined. Neither is anything mentioned about clinical appearance of hypothyroid symptoms in connections with laboratory measures before considering levothyroxine supplementation.
Many of the stigmatas in the Introductions, like on page 2 about the HDL seem odd in this context and is not necessary.
Sometimes the authors use Turner syndrome and sometimes TS. Do not write TS patients. It should be women or patients with Turner syndrome.
Hypothyroidism should be included in the search key words
One sentence after GH treatment on page 5 is not easy to understand. It starts with: “TT4 as well as fT4 tended to decrease, increase in TT3serum levels, TSH serum levels were significantly increased, serum TSH values did not significantly differ, increased serum IGF-I levels and bone resorption and formation serum markers, educed serum SHBG levels.” What will the authors want to say by this? TSH increased but in the same sentence TSH did not differ significantly.
Figure 1 The Figure legend should be more informative. For example, the flow chart in Figure 1. There is no information if this is about Turner syndrome or which years the search was framed on.
Figure 2 the year for each publication should be included after the Authors´ name. Abbreviation in the legends should be avoided.
Figure 2 should have the same lay-out as Table 1 with the authors in the same order with year added vertical in the left panel and the variables horizontally.
The text on page 6 is irrelevant for the present topic of thyroid disease.
The chapter 3.2 on page 7 is relevant. However, the authors introduce TS as a short for Turner syndrome but do not use it after some sentences.
Chapter 3.2.1 contains several sentences starting with “We”. Please avoid this and write in third person please.
Table 2 should be changed in lay-out similar as mentioned for Table 1 above. Legends must be more informative.
Page 8 first line higher prevalence that what?
Table 5. What does the Table show? Subjects?
Chapter 3.3 is irrelevant for the women with Turner syndrome in this review. Free T4 has not shown to be affected in Turner syndrome.
Chapter 3.3.1 Definition of hypothyroidism is needed. What is considered hypothyroidism and why should laboratory levels be treated only? The guidelines in Turner syndrome in 2024 and previous years (Saenger et al 2001, Bondy et al 2007, Gravholt et al 2017) recommend evaluation of thyroid function every year in the adult women with Turner syndrome.
There is nothing in the entire text of the manuscript written about concomitant symptoms of suspected hypothyroidism.
Page 13 Placebo-controlled trials are recommended. Of what therapy and what outcome is meant? Further down on page 13 randomized trials on genetics are recommended. How possible or feasible is this to be performed in Turner syndrome? Several studies in this patient group have not shown any predominant genetic group for hypothyroidism. Hypothyroidism has been evenly distributed irrespective of monosomy or mosaicism in Turner syndrome, respectively. Last lines; what do the author base the postulation on, or the evidence for, that treating hypothyroidism in Turner syndrome should lead to prolonged life and improved quality of life?
References should be uniform regarding pages. Some references lack pages and some have 325-66 written followed by some with all pages mentioned; 986-993 for example.
Some misprints of wordings throughout the manuscript.
Comments on the Quality of English LanguageCorrection for English language and spelling is recommended.
Author Response
Dear Reviewer,
The authors thank you very much for the effort put into the review and for valuable comments. The comments helped us significantly improve the manuscript.
We have made corrections to the manuscript based on your comments. Each comment was considered individually and we have made appropriate changes to our manuscript. The manuscript has been revised by an English language expert, and the relevant document confirming the review and proofreading is attached as an additional document.
All answers are presented in the file below with our answers. Once again, we sincerely thank you for your contribution, which allowed us to improve the quality of our manuscript.
Yours Sincerely,
The authors

Reviewer 2 Report
Comments and Suggestions for Authors
Thank you very much for inviting me to review the following review article. I believe that the work is of a generally high level and contains a number of valuable contents.
The described process of selecting studies for analysis seems to be correct. The small number of studies meeting the inclusion criteria is due to the explicit recruitment criteria chosen.
There are several issues that could be improved to increase the value of this manuscript.
1. The introduction contains a little too much general information about Turner syndrome, mainly concerning phenotypic features. I would suggest that the problem of autoimmunity in this syndrome be highlighted more in this section.
2. In results: In all the tables in the first box is written: “author and year” while the year of publication has been omitted each time.
3. Sect. 3.3 "Treatment" does not accurately describe the problem addressed in the topic of the paper. I would suggest changing the subtitle to "screeninig and treatment." It should be mentioned that the treatment goals for autoimmune disease/thyroid dysfunction in people with TS are not different from other populations. I suggest describing the problem of thyroid treatment first and only then the interplay of GH and thyroid setup.
Author Response

(The authors gave the same response as above.)
